# The Physics behind the Modulation of Thermionic Current in Photodetectors Based on Graphene Embedded between Amorphous and Crystalline Silicon

**DOI:** 10.3390/nano13050872

**Published:** 2023-02-26

**Authors:** Teresa Crisci, Piera Maccagnani, Luigi Moretti, Caterina Summonte, Mariano Gioffrè, Rita Rizzoli, Maurizio Casalino

**Affiliations:** 1Institute of Applied Science and Intelligent Systems “Eduardo Caianiello” (CNR), 80131 Napoli, Italy; 2Department of Mathematics and Physics, University of Campania “Luigi Vanvitelli”, 81100 Caserta, Italy; 3Institute for Microelectronics and Microsystems (CNR), 40129 Bologna, Italy; 4Department of Physics and Earth Sciences, University of Ferrara, Via Giuseppe Saragat 1/c, 44122 Ferrara, Italy

**Keywords:** graphene, photodetector, near infrared, encapsulation, silicon photonics

## Abstract

In this work, we investigate a vertically illuminated near-infrared photodetector based on a graphene layer physically embedded between a crystalline and a hydrogenated silicon layer. Under near-infrared illumination, our devices show an unforeseen increase in the thermionic current. This effect has been ascribed to the lowering of the graphene/crystalline silicon Schottky barrier as the result of an upward shift in the graphene Fermi level induced by the charge carriers released from traps localized at the graphene/amorphous silicon interface under illumination. A complex model reproducing the experimental observations has been presented and discussed. Responsivity of our devices exhibits a maximum value of 27 mA/W at 1543 nm under an optical power of 8.7 μW, which could be further improved at lower optical power. Our findings offer new insights, highlighting at the same time a new detection mechanism which could be exploited for developing near-infrared silicon photodetectors suitable for power monitoring applications.

## 1. Introduction

Silicon (Si) is a well-known semiconductor and its properties have been extensively studied as a result of the expansion of the microelectronic industry. Unfortunately, the indirect nature and the size of its bandgap preclude its use in the realization of photonic devices such as lasers and near-infrared (NIR) photodetectors (PDs) [1].

Graphene and graphene-related materials have been employed in many fields. For example, magnetic graphene oxide/poly(vinylalcohol) (PVA) composite gels have shown enhanced adsorption for environmental clean-up combined with a convenient magnetic separation capability [2], while graphene/carbon nanotube 3D nanocomposites have shown the capability to selectively detect tetrabromobisphenols [3], which has received huge attention due to its environmental toxicity. In addition, graphene-based three-band adjustable perfect absorbers have been investigated and show potential in practical sensing applications [4]. Moreover, graphene and graphene-like materials have recently shown great potential in photonic and optoelectronic applications [5,6]. Graphene (Gr) integrated on Si has recently opened intriguing new perspectives in the field of NIR photodetection by taking advantage of two detection mechanisms: the photogating effect [5] and the internal photoemission effect (IPE) [7].

The former approach is based on the trapping of photogenerated carriers in localized states, generating a local voltage that can modulate the Gr channel conductance, typically within FET structures. Hybrid phototransistors based on PbS quantum dots (QDs) distributed on Gr layers transferred on top of SiO_2_/Si substrates have shown an impressive gain of 10^8^ electrons per photon at 600 nm due to the long lifetime of carriers trapped in PbS QDs, i.e., to the large amount of charge carriers that must flow through the Gr channel to re-establish charge neutrality [8]. By employing a similar approach, Goossens et al. have reported a short-wave infrared (SWIR) image sensor based on the integration of a Gr layer decorated by PbS QDs with a Si read-out circuit, demonstrating the first impressive convergence between QDs, Gr, and CMOS electronic circuitry [9]. Gr employed as a channel in a hybrid junction formed with MoS_2_ has also achieved a photo-gain exceeding 10^8^ [10,11], while nanotube-Gr hybrid junctions have shown a broadband detection (from 200 to 1550 nm), a response time of 100 μs, and a photo-gain of approximately 10^5^ [12]. In addition, a responsivity of more than 1 A/W in a spectrum ranging from 1.3–3.2 μm has been demonstrated in PDs based on two layers of Gr separated by a thin TaO_5_ layer [13]. The IPE approach has been oriented towards the development of cryogenic CCD image sensors operating in the SWIR range. IPE concerns the photoexcitation of charge carriers in the metal/silicide region that are emitted into Si after overcoming the junction Schottky barrier [7,14]. A focal plane array (FPA) of 512 × 512 PtSi/Si Schottky PDs operating from 3 to 5 μm has been manufactured, proving the possible convergence between photonics and electronics in the same Si substrate [15]. Unfortunately, these devices were characterized by a poor external quantum efficiency (<1%), leading to the investigations of many strategies for improving it. They include the use of gratings [16], antennas [17], surface plasmon polaritons (SPPs) [18], and Si nanoparticles (NPs) [19]. Gr/Si Schottky PDs operating in the NIR regime have recently shown higher efficiencies with respect to the junctions based on metals or silicides. The origin of this enhancement has been ascribed to the two-dimensional nature of Gr and to the increased probability that photoexcited charge carriers are emitted into Si [20,21]. Unfortunately, the poor optical absorption of Gr (2.3%) is not conducive to achieving high-efficiency PDs, therefore many approaches, mainly based on waveguiding structures, have been proposed [22,23]. Free-space Gr/Si PDs based on a Fabry–Pérot optical microcavity have also been theoretically proposed for increasing the Gr optical absorption [24,25]. In ref. [25] a Gr layer deposited on the edge of a Fabry–Pérot optical microcavity has provided an increased Gr optical absorption of approximately 8% and a responsivity of 20 mA/W at 1550 nm and with 10 V of reverse bias applied. On the other hand, in ref. [24] we have theoretically demonstrated the advantage of placing a Gr layer in the middle of a Si-based optical microcavity by leveraging the amorphous hydrogenated silicon (a-Si:H). Indeed, this material presents two aspects of paramount importance: the complex refractive index very similar to that of crystalline silicon (c-Si) and the manufacturing process that can be carried out at low temperatures by plasma-enhanced chemical vapor deposition (PECVD), avoiding the damage that would otherwise be induced in Gr during its deposition. In order to verify the theoretical predictions of ref. [24], we fabricated a novel metal–semiconductor–metal (MSM) a-Si:H/Gr/c-Si novel PD and observed an unexpected modulation of thermionic current under illumination, which could be exploited for detecting the NIR radiation.

The vertically illuminated NIR MSM photodetector proposed is based on the modulation of the thermionic current of a Gr/c-Si Schottky junction. The structure is a low-finesse Fabry-Pérot optical microcavity with a few layers Gr (FLG) embedded between the crystalline top-silicon of a silicon-on-insulator (SOI) substrate and an a-Si:H layer deposited at low temperature in a PECVD system. The structure has been designed to place the trap levels of the a-Si:H/Gr interface where the optical field of approximately 1550 nm has the maximum intensity inside the cavity. We have discovered that under illumination, the trapped charges localized at the interface with a-Si:H are released into Gr providing an upward shift of its Fermi level and, consequently, a reduction in the Schottky Gr/c-Si barrier, detectable by measuring the thermionic current. The physics behind this effect has been fully understood and elucidated. For these devices, a maximum responsivity of 27 mA/W at 1543 nm under an optical power of 8.7 μW has been reported. This work investigates a novel detection mechanism advantageous in the development of near-infrared silicon photodetectors useful for power monitoring applications.

## 2. Device Concept and Design

The proposed PD is based on an a-Si:H/Gr/c-Si/Al hybrid structure, manufactured starting from an SOI substrate as shown in Figure 1a (the detailed description of the fabrication process is reported in Section 3.1). The Gr/c-Si/Al junction represents an MSM structure, the band diagram for which is shown in Figure 1b, depicting the condition in which a positive voltage higher than flat-band voltage is applied to Al with respect to Gr. It is worth noting that Figure 1b reports the band diagram of the MSM junction for a single layer Gr, although it is very well known and widely discussed in the literature that a few layers of Gr are characterized by a finite effective mass m* and are thus described by hyperbolic bands [26,27,28]. However, for the sake of simplicity and ease of interpretation, and given that the theory used in this research is generally considered valid, we have depicted the case of an MSM junction with a single-layer Gr and linear energetic bands.

The detection mechanism exploits the defects at the a-Si:H/Gr interface. Under near-infrared illumination, the traps shed charges in Gr causing a change of the Fermi level and the Schottky barrier. Consequently, a different thermionic current is observed flowing through the Gr/c-Si Schottky junction as shown in the band diagram of Figure 1b. The structure is essentially a Fabry–Pérot microcavity where the top and bottom mirrors of the cavity are provided by the air/a-Si:H and c-Si/silicon dioxide (SiO_2_) interfaces, respectively. The a-Si:H thickness has been designed to set the maximum amplitude of the resonant mode at 1550 nm around the Gr/a-Si:H interface.

An extensive study of high-finesse resonant optical cavities applied to the detector has been described in refs. [24,29]. The photodetector was fabricated using an SOI substrate with a 220 nm thick c-Si top layer.

To optimize the Fabry–Pérot microcavity, we calculated that if the refractive indexes of the c-Si and a-Si:H layers are, respectively, 3.48 [30] and 3.58 [29] at 1550 nm, for a graphene thickness of 1.34 nm (which nominally corresponds to a thickness of 4 layers), an a-Si:H thickness of 208 nm allows us to localize the maximum of the standing wave arising in the cavity at the a-Si:H/graphene interface, as shown in Figure 2a. The corresponding Gr spectral optical absorption at the wavelength of interest as a function of the a-Si:H thickness is reported in Figure 2b, while in the inset the change in Gr optical absorption as a function of different wavelengths and for various a-Si:H thicknesses is shown. Both optical field distribution and spectral optical absorption of Figure 2 have been calculated by means of the generalized scattering matrix method using the code ‘Optical’ [31] with the complex refractive index dispersion of all involved materials. In particular, the refractive index of Gr (*n_g_*) has been calculated through the following expression [32]:(1)$$n_{g}=\sqrt{\varepsilon_{g}}=\sqrt{5.7+j\frac{\alpha \lambda }{2d_{g}}}$$
where λ is the wavelength, *d_g_* = 0.335 nm is the monolayer Gr thickness, *ε_g_* is the relative permittivity of Gr, and $\alpha =\frac{1}{4{\pi \varepsilon }_{0}}\cdot \frac{q^{2}}{\hbar c}=0.0073$ is the fine structure constant [33] in SI base units (where *ε*_0_ = 8.854 × 10^−12^ F/m is the vacuum permittivity, *c* = 3 × 10^8^ m/s is the speed of light in a vacuum, and ħ = 1.055 × 10^−34^ J·s is the reduced Planck constant). By Equation (1), the complex refractive index of Gr is *n_g_* = 3.43 − j2.46 at 1550 nm. In the Appendix A, we have verified that Equation (1) can also be applied in the case of multiple layers (Appendix A).

## 3. Materials and Methods

### 3.1. Device Fabrication

Devices were fabricated starting with an SOI wafer with a P-type lightly-doped (14–22 Ω·cm) Si top layer and a 3 μm-thick buried silicon oxide (BOX) layer, as shown in Figure 1a. Firstly, a 120 nm-thick TEOS oxide was deposited on the Si top layer to work as an insulating layer. The collecting electrode was defined by optical lithography followed by oxide wet etching in buffered HF solution (buffered oxide etching, BOE), thermal evaporation of Au/Cr/Al (70 nm/10 nm/70 nm), and the lift-off process. After the lithographic definition of the device active areas through the SiO_2_ etching in standard BOE solution, the substrates were ready for Gr transfer. FLG films were grown on 25 μm-thick copper foils (purchased from Alfa Aesar, Kandel, Germany) by means of Catalytic-CVD (C-CVD) [34]. The Cu foil was annealed at 1000 °C for 30 min with a 50 sccm H_2_ flow, then the carbon precursor CH_4_ (50 sccm) was introduced into the CVD for 10 min while the H_2_ flow was increased to 500 sccm to be in the range where the size of the Cu grains and the Gr domain boundaries are hundreds of microns. At the end of this process, Gr was covered with 1 μm thick poly-methyl-methacrylate polymer (PMMA 950-A7, micro resist technology GmbH, Berlin, Germany). PMMA is used as supporting layer during the etching of Cu in the APS solution (ammonium persulfate, 50 g/L in water). Before transferring the Gr onto the SOI substrate, the silicon was dipped in a solution of hydrofluoric acid (HF in water 1:1000) for 30 s to remove native oxide from diode active areas. Then, PMMA was removed in acetone vapors and Gr was lithographically patterned using HPR-504 resist, and dry etched in an oxygen plasma (25% O_2_ + 75% N_2_). The average thickness of the Gr film determined using AFM in tapping mode was in a range from 1.9 to 2.4 nm. Therefore, by considering that STM measurements in ultrahigh vacuum conditions have reported a monolayer Gr thickness of 0.42 nm, and that the thickness of a single-layer Gr obtained with AFM is 0.6 ± 0.2 nm [35], a few layers of Gr (approximately 4–5 layers), can be estimated. Metal electrodes on Gr were realized by thermal evaporation of Cr/Au (10–70 nm) after a photolithographic step where PMMA (exposed in DUV light, 248 nm) was used as the resist. Finally, an amorphous silicon (a-Si:H) layer was deposited on top of the chip using a PECVD system, compatible with the back end of line (BEOL) semiconductor device fabrication thanks to its low thermal budget. The deposition was performed at 13.56 MHz by means of a SiH_4_ plasma at a temperature of 170 °C. Finally, a-Si:H was patterned and dry etched in an SF_6_/O_2_ plasma. The scheme of the fabrication procedure here described is reported in Appendix A, and a top view of the device sketched in Figure 1a is reported in Figure 3a, taken by an optical microscope.

### 3.2. Raman Analysis

The Raman spectra of graphene were acquired using a Renishaw InVia micro-spectrometer working at 10% power with a spot ∼2 μm width and 25 μm length and a 50× objective. The excitation energy was 785 nm with a spectral resolution of 2 cm^−1^.

### 3.3. Electrical Characterization

Current–voltage measurements on the photodetectors were acquired using a source meter (Keysight B2902A, Santa Rosa, CA, USA) connected to a PC and driven by custom codes written in Matlab (The MathWorks Inc., Natick, MA, USA). Samples were placed on the holder of a probe station and the 2 electrodes were connected by 2 tips of XYZ micromanipulators (450/360MT-6 and 550/360MT-6 series, The Micromanipulator Co., Carson City, NV, USA). Each I–V curve is the average of 5 measurements performed by varying the voltages repetitively from −5 to 5 V.

### 3.4. Electro-Optical Characterization

Responsivity measurements were performed using a CW laser at 1543 nm (ANDO AQ4321D, Tokyo, JP) and measuring the I–V curve of the device using a source meter (Agilent B2902A) both in dark conditions (*i_D_*) and under illumination (*i_L_*). *i_D_* and *i_L_* were measured alternatively 10 times and then the net current *i_Ph_* = *i_L_* − *i_D_* was calculated as the average value. Samples were placed on the holder of a probe station and the 2 electrodes were contacted using 2 tips of XYZ micromanipulators (450/360MT-6 and 550/360MT-6 series, The Micromanipulator Co., Carson City, NV, USA). The device was illuminated from the a-Si:H side and the NIR beam was aligned to the active area using an IR microscope equipped with an IR CCD. The incident optical power P was separately measured by a commercial calibrated InGaAs PD and normalized to the active area of the device under test. Finally, the responsivity was calculated as *R* = *i_Ph_*/P. All measurements were performed after maintaining the device at the fixed voltage of −21 V for 20 min to minimize the dark current. NIR illumination was applied for 1 s in all measurements. Experimental results are reported in Figure 5a–c.

## 4. Experimental Results

### 4.1. Raman Measurements

During the manufacturing process the graphene layer is exposed to plasma in the a-Si:H deposition process (PECVD system), so damage to the lattice could arise and cause a worsening of the structural properties of the material due to the ion bombardment [36]. To investigate this issue, the Gr quality was monitored through Raman measurements. Indeed, there exists a strong correlation between the intensity of the defect-related Raman D-band and the energy of impinging ions, as demonstrated by Ahlberg et al. in [37]. The Raman spectra of graphene were acquired before and after the deposition of the a-Si:H deposition process (PECVD system) and the results are shown in Figure 3b. Since the peaks of graphene on flat silicon are typically difficult to detect in standard Raman investigation [38], the spectra were obtained for Gr on SiO_2_ in the region immediately outside the active diode area.

The characteristic phonon modes are well observed for uncapped graphene (blue spectrum in Figure 3b), while for graphene capped with a-Si:H, a pronounced decrease in the intensity of the graphene-related phonon modes can be noticed. This behavior was ascribed to the optical absorption of the a-Si:H layer through which the phonon modes of graphene were measured (red spectrum in Figure 3b). The Raman spectrum for bare graphene shows a small D band at 1312 cm^−1^ related to the transfer and patterning of Gr, and two strong peaks at 1592 cm^−1^ and 2623 cm^−1^ assigned to the G and 2D bands, respectively. The deposition and patterning of the a-Si:H layer introduces the broad peak centered at 2011 cm^−1^ associated with Si-H bond vibrations, but the characteristic Raman spectrum of Gr is preserved and only a slight shift in the frequencies (in the order of spectral resolution) for D, G, and 2D modes was detected (Table 1).

The ratio *I_D_*/*I_G_* allows the level of induced damages to be estimated [37,39]. In particular, in the case of large graphene crystals (>>30 nm), the mean distance between defects *L_D_* can be calculated using a simplified equation, valid for visible range excitation: $\frac{I_{D}}{I_{G}}=\left(1.8\pm 0.5\right)\times 10^{-9}\lambda_{R}^{4}/L_{D}^{2}$, where $\lambda_{R}$ = 785 nm is the laser wavelength used for micro-Raman measurements. We obtained *L_D_* = 48 nm for bare graphene, while a slight reduction down to 39 nm for graphene capped with a-Si:H was observed.

### 4.2. Electrical Measurements

Current–voltage measurements were performed by applying a voltage bias (positive or negative) to the Gr contact, while silicon was kept grounded. The experimental results observed on the photodetector at room temperature under dark conditions are reported in Figure 4. The time dependence of the measured current applying a fixed voltage of −21 V is shown in Figure 4b. This time dependence can be linked to the process of charge trapping mentioned above and will be discussed in the next section. Information on carrier density in the graphene layer (useful for the following discussion) has been obtained through the van der Pauw–Hall measurement, a common technique for the characterization of thin films. In ambient conditions, the average value obtained for as-transferred graphene indicates a strong P-type doping of 9 × 10^12^ cm^−2^, likely due to adsorbed O_2_, moisture, and polymer contaminants. As a result of the a-Si:H deposition, there is a reduction in P-type doping in capped graphene and hole concentration decreases to 3.5 × 10^12^ cm^−2^, moving the graphene Fermi level upwards towards the Dirac point.

### 4.3. Photogenerated Current and Responsivity Measurements

Responsivity (the ratio between the measured current and the incident optical power) is a parameter used to quantify the device efficiency. For comparison, the responsivity measured at the Gr/c-Si Schottky junction before the a-Si:H deposition is reported in Figure 5d.

The responsivity as a function of the incident optical power shown in Figure 6 was derived from Figure 5c at −21 V. In the inset, the efficiency-lifetime carrier product is reported, and will be discussed in the next section.

## 5. Discussion

### 5.1. Discussion of the Electrical Results

The I–V curve shown in Figure 4a shows the typical behavior of a metal–semiconductor–metal structure. This MSM is constituted by two Schottky junctions: Gr/c-Si and Al/c-Si, respectively. By applying a continuous external voltage to the device, one junction is forward biased while the other is reverse biased. At the so-called flat-band voltage $V_{FB}=(qN/2\varepsilon )L^{2}$ (where *q* is the electron charge, *N* the semiconductor doping, *ε* the silicon dielectric constant, and *L* the distance between the two electrodes), the c-Si semiconductor is fully depleted, energy bands become flat in correspondence with the reverse-biased junction, and the total current that flows through the device can be written as the sum of electrons injected by the forward-biased junction and holes injected by the reverse-biased junction [40]. In Figure 4a, the negative axis corresponds to a negative voltage applied to the Gr contact with respect to the grounded Al contact, while the positive axis corresponds to a positive voltage applied to the Gr contact with respect to the grounded Al contact. As shown in Figure 4b, these devices show a dependence in time of the current under a fixed voltage of −21 V that was not observed on the same devices prior to the a-Si:H deposition. The time dependence shown in Figure 4b must be ascribed to the presence of parasitic capacitance originating from the deposition of a-Si:H and to the presence of interface traps at the a-Si:H/Gr interface. In other words, the charge carriers are trapped in these defects, giving rise to a capacitance schematized as *C_it_* in the electrical circuit of Figure 7. A further effect of the presence of interface traps at the a-Si:H/Gr interface is the increase in the Gr resistance attributed to extrinsic scattering mechanisms, as widely reported in the literature for the top-gated Gr field effect transistor (G-FET) [41]. The Gr resistance has been schematized as *R_it_* in the electrical circuit of Figure 7. Moreover, under the negative biasing conditions (the Gr/c-Si and the Al/c-Si junctions are forward- and reverse biased, respectively) after experiencing the interaction with interfacial traps, the charge carriers are injected over the Gr/c-Si Schottky barrier (which can be represented as a resistance *R_j_*), and after passing through the resistance *R_s_* of the Si substrate, they can be collected by the Al contact. The resistance *R*_0_ schematized in Figure 7 is the sum of these two resistances (*R*_0_ = *R_j_* + *R_s_*). For a better understanding of the electrical circuit in Figure 7, two sketches of the device showing both the path of carriers moving from the graphene contact towards the aluminum electrode and the aforementioned electronic components are reported in the Appendix A.

Finally, under NIR illumination on the Gr/c-Si junction, we observed an additive current *i_ph_* that can be represented as a current generator operating in parallel to the junction schematized as *R*_0_ in Figure 7.

The circuit shown in Figure 7 can be described by the following first order differential equation:(2)$$\frac{dv_{c}}{dt}+\frac{v_{c}}{\left(R_{it}\backslash \backslash R_{0}\right){\cdot C}_{it}}=\frac{V_{i}}{R_{0}C_{it}}+\frac{i_{Ph}}{C_{it}}$$
whose solution is the following:(3)$$v_{c}\left(t\right)=e^{-\frac{t}{\tau_{it}}}\cdot \left[v_{c}\left(0\right)+\int_{0}^{t}\left(\frac{V_{i}}{R_{0}C_{it}}+\frac{i_{Ph}}{C_{it}}\right)\cdot e^{\frac{s}{\tau_{it}}}ds\right]$$
where *V_i_* is the constant bias applied to the device at the initial time *t* = 0, *v_c_* is the voltage drop on the capacitance *C_it_* in parallel with the resistance *R_it_*, *v*_0_ is the voltage drop on the resistance *R*_0_, while the constant time was defined as *τ_it_* = (*R_it_*\\*R*_0_)*C_it_*.

If the device is NIR-illuminated starting from time *t*_0_, the photogenerated current can be written as *I_Ph_*(*t*) = *I_Ph_ u*(*t* − *t*_0_), where *I_Ph_* is the intensity of the current originating by a continuous optical power impinging on the Gr/c-Si junction, while *u*(*t*) is the step function. Then, in this condition:(4)$$v_{c}\left(t\right)=v_{c}\left(0\right)e^{-\frac{t}{\tau_{it}}}+\frac{V_{i}\tau_{it}}{R_{0}C_{it}}\cdot \left(1-e^{-\frac{t}{\tau_{it}}}\right)+\frac{I_{Ph}\tau_{it}}{C_{it}}\cdot \left(1-e^{-\frac{t-t_{0}}{\tau_{it}}}\right)\cdot u\left(t-t_{0}\right)$$
where *v_c_*(0) is the voltage drop on the capacitance at the initial instant *t* = 0. Therefore, the current that flows through the resistance *R*_0_ can be written as:(5)$$i_{0}=\frac{V_{i}-v_{c}}{R_{0}}=\frac{V_{i}}{R_{it}+R_{0}}\cdot \left({1+\frac{R_{it}}{R_{0}}\cdot e}^{-\frac{t}{\tau_{it}}}\right)-\frac{I_{Ph}R_{it}}{{R_{0}+R}_{it}}\cdot \left(1-e^{-\frac{t-t_{0}}{\tau_{it}}}\right)\cdot u\left(t-t_{0}\right)-\frac{v_{c}\left(0\right)}{R_{0}}e^{-\frac{t}{\tau_{it}}}$$

The current *i_M_* = *I_Ph_* + *i*_0_ (the current experimentally measured under NIR illumination) can therefore be written as:(6)$$i_{M}=\left[I_{Ph}\left(1-\frac{R_{it}}{{R_{0}+R}_{it}}\cdot \left(1-e^{-\frac{t-t_{0}}{\tau_{it}}}\right)\right)\cdot u\left(t-t_{0}\right)\right]+\left[\frac{V_{i}}{R_{it}+R_{0}}\cdot \left({1+\frac{R_{it}}{R_{0}}\cdot e}^{-\frac{t}{\tau_{it}}}\right)-\frac{v_{c}\left(0\right)}{R_{0}}e^{-\frac{t}{\tau_{it}}}\right]=i_{L}+i_{D}$$

It is worth observing that the measured current *i_M_* is composed of two contributions, both depending on time and enclosed in the square brackets of Equation (6): the first contribution (*i_L_*) is due to the light, while the second contribution (*i_D_*) can be considered as a dark current, since it does not depend on the photogenerated current. By considering a completely discharged capacitance at *t* = 0 (*v_c_*(0) = 0), the measured current *i_M_* reduces to the dark current *i_D_*:(7)$$i_{D}=\frac{V_{i}}{R_{it}+R_{0}}\cdot \left({1+\frac{R_{it}}{R_{0}}\cdot e}^{-\frac{t}{C_{it}\left(R_{0}//R_{it}\right)}}\right)$$
which depends on time, as shown in Figure 4b, and reduces after a transient *t* >> *τ_it_* to the minimum constant value *i_D_* = *V_i_*/(*R_it_* + *R*_0_). By taking advantage of Equation (7), it is also possible to extract *R_it_*, *R*_0_, and *C_it_* by a fitting procedure (R-square of 0.93) on the experimental data, as shown in Figure 4b. The resulting fitting parameters are: *R*_0_ = 5.1 MΩ, *R_it_* = 15.1 MΩ and *C_it_* = 14.3 μF. By considering the active area of the Gr/c-Si junction (A_Gr_) of 1.26 × 10^−3^ cm^2^, the capacitance due to the traps per unit of area becomes 11.35 mF/cm^2^. The results of the fitting parameters can be used to evaluate the transitory as well as the total charge trapped in this time. Indeed, the time constant *τ_it_* = *C_it_* (*R_it_*//*R*_0_) can be calculated as 54 s.

As for the contribution of the measured current during NIR illumination, when *t* ≥ *t*_0_, a current contribution due to the light *i_L_* is added to the dark current:(8)$$i_{L}=\left[I_{Ph}\left(1-\frac{R_{it}}{{R_{0}+R}_{it}}\cdot \left(1-e^{-\frac{t-t_{0}}{\tau_{it}}}\right)\right)\cdot u\left(t-t_{0}\right)\right]$$

If we consider *t_M_* = *t* − *t*_0_ as the time during which the optical power is applied on the Gr/c-Si junction and *t_M_* << *τ_it_* = 54 s, Equation (8) reduces to:(9)$$i_{L}=I_{Ph}\cdot \left(1-\frac{t_{M}}{{R_{0}C}_{it}}\right)≅I_{Ph}$$

It is worth noting that *R*_0_*C_it_* > *C_it_*(*R*_0_\\*R_it_*). Therefore, if the measurement time *t_M_* << *τ_it_*, the condition *t_M_* << *R*_0_*C_it_* is certainly fulfilled and *i_L_* approaches I_Ph_, as reported in Equation (9). Measurements reported in Figure 5a,b were performed by maintaining the device at *V_i_* = −21 V for 20 min before applying an NIR illumination for one second, alternating between dark and light 10 times. In this condition, the current measured under NIR illumination is *I_Ph_* + *V_i_*/(*R_it_* + *R*_0_) while the current in dark conditions is *V_i_*/(*R_it_* + *R*_0_), and the difference between these two currents gives the photogenerated current *I_Ph_*, useful for the evaluation of responsivity.

### 5.2. Discussion on the Electro-Optical Results and Physical Interpretation

The measured responsivities shown in Figure 5c raise the question of which photoconversion mechanism is involved. We cannot employ the photogating theory because it involves FET structures and not Gr/c-Si Schottky junctions [5]. These, on the other hand, can be associated with a detection mechanism based on an internal photoemission effect [7]. However, the measurements carried out on the Gr/c-Si Schottky junction before the a-Si:H deposition (reported in Figure 5d) showed a limited responsivity of only 0.06 mA/W at 1550 nm and −20 V and no dependence on the incident optical power as predicted by the IPE theory [7]. With the addition of a-Si:H, the device responsivity increases by more than one order of magnitude, evidencing the clear dependence on the incident optical power shown in Figure 5c which is typically associated with the presence of traps [5]. In our opinion, the traps at the a-Si:H/Gr interface play a key role in the photoconversion mechanism, the theoretical model of which will be derived in this work and compared with the experimental data. As shown in Figure 5a,b, by applying a negative voltage on Gr (Al is grounded) under NIR illumination we observe a large increase in the photodetector current. This increase could be explained by considering an upward shift in the Gr Fermi level, i.e., a lowering of the Gr/c-Si Schottky barrier as shown in the band diagram of Figure 1b. Furthermore, very significantly, the photogenerated current increases its value with respect to the dark current by translating rigidly downwards, as shown in Figure 5a,b. This remark suggests that the photogenerated current is nothing more than a thermionic current flowing through a lower Gr/c-Si Schottky barrier. The Gr/c-Si Schottky barrier reduction could be attributed to the electrons trapped in the defects at the a-Si:H/Gr interface which are released into Gr under illumination. With this line of reasoning, we determine how the thermionic current supported by the device changes as it passes from dark conditions to NIR illumination. In the following theoretical discussion, we consider a single layer of graphene. Indeed, since the focus of this paper is the elaboration of a theory able to explain the new physical phenomena observed, we treat the case of a single layer for which well-established closed analytical formulas exist. As a consequence, the derived expressions are precise in the case of a single layer of graphene, while they can be considered an approximation for several layers of graphene. In any case, the reasoning can be generalized and extended to the case of N layers of graphene if the expression for the Fermi energy level as a function of the carrier density is known. 

In our model we assume that: (a) the dark current is primarily due to thermionic emission, (b) the applied bias is greater than the flat-band voltage, and (c) recombination in the space charge region, breakdown effects, and surface state transport can be neglected. If a negative voltage is applied to the Gr contact with respect to the grounded Al contact, the Gr/p-Si junction is forward biased, whereas the Al/p-Si junction is reverse biased providing the energy band diagram shown in Figure 1b under dark conditions. As shown in Figure 1b, the current flowing through the device before illumination (*I_TD_*) is the sum of the dark current, due to thermionic emission of electrons overcoming the potential barrier (*ϕ**^Gr^_B_*) from the Gr contact ($I_{N,dark}^{Gr}$), and the thermionic emission of holes, overcoming the potential barrier (*ϕ**^Al^_B_*) from the Al contact (*I_P_^Al^*):(10)$$I_{TD}={I_{N,dark}^{Gr}+I}_{P}^{Al}={A_{Gr}A_{N}^{*}T^{2}e^{-\frac{{q\phi }_{B}^{Gr}}{kT}}+A}_{Al}A_{P}^{*}T^{2}e^{-\frac{{q\phi }_{B}^{Al}}{kT}}$$
where *A_Gr_* and *A_Al_* are the *Gr* and *Al* in contact with Si, respectively, *T* is the absolute temperature, *k* is the Boltzmann constant, and *A*_P_* and *A*_N_* are the Richardson constant for P-type and N-type Si, respectively. Indeed, as reported in ref. [40], the current continuity requirement dictates that the total current in the MSM structure must equal the sum of the thermionic current which flows through the two junctions.

Under NIR illumination of the Gr-active area, the increase in total current flowing through the device (*I_TL_*) suggests a reduction $∆\phi_{B}$ in the Gr/c-Si Schottky barrier:(11)$$I_{TL}={I_{N,light}^{Gr}+I}_{P}^{Al}=A_{Gr}A_{N}^{*}T^{2}e^{-\frac{q\left(\phi_{B}^{Gr}-∆\phi_{B}\right)}{kT}}+A_{Al}A_{P}^{*}T^{2}e^{-\frac{q\phi_{B}^{Al}}{kT}}$$

If the photogenerated current is defined as *I_Ph_* = *I_TL_* − *I_TD_*, we can write:(12)$$I_{Ph}=A_{Gr}A_{N}^{*}T^{2}e^{-\frac{{q\phi }_{B}^{Gr}}{kT}}\left(e^{\frac{q∆\phi_{B}}{kT}}-1\right)$$

As mentioned above, the reduction in the Gr/c-Si Schottky barrier can be ascribed to an upward shift in the Gr Fermi level towards the Dirac point, which depends on the charges stored in the Gr layer [42,43]:(13)$$∆E_{F}^{D}=E_{F}^{D}-E_{F0}^{}=-sign\left(n_{0}+\sqrt{\frac{2\epsilon }{q}N\left(V_{bi}-V_{F}\right)}\right)\hbar v_{F}\sqrt{\pi \left|n_{0}+\sqrt{\frac{2\epsilon }{q}N\left(V_{bi}-V_{F}\right)}\right|}$$
where *E_F_*_0_ is the Dirac point, $E_{F}^{D}$ the Gr Fermi level, *N* is the doping of the crystalline silicon substrate, $v_{F}$ is the Fermi velocity, *V_bi_* is the built-in potential, *V_F_* is the voltage across the Gr/c-Si junction, *n*_0_ is the capped Gr doping, while *sign* is the signum mathematical function. When the Al contact is strongly positively polarized with respect to the Gr contact, the Gr/c-Si junction is forward biased and the external voltage *V_F_* dropping across the Gr/c-Si junction tends to approach to the built-in voltage of the Gr/c-Si junction (*V_F_* 🡪 *V_bi_*); in other words, only a small depletion region is due to the Gr/c-Si junction. Thus, the term $\sqrt{\frac{2\varepsilon }{q}N\left(V_{bi}-V_{F}\right)}$ can be neglected, providing $\Delta E_{F}^{D}=-sign\left(n_{o}\right)\cdot \hbar v_{F}\sqrt{\pi \left|n_{o}\right|}$.

Under illumination, the a-Si:H/Gr interface traps release some charges per unit of area (*N_c_*) into Gr whcih are added to *n*_0_, therefore shifting the Gr Fermi level by a $∆E_{F}^{L}$ amount up to the level $E_{F}^{L}$:(14)$$∆E_{F}^{L}=E_{F}^{L}-E_{F0}^{}=-sign\left(n_{0}+N_{c}\right)\hbar v_{F}\sqrt{\pi \left|n_{0}+N_{c}\right|}$$

Therefore, the change in Schottky barrier Δ*ϕ*_B_ under illumination can be defined as:(15)$$q∆\phi_{B}={∆E}_{F}^{L}-{∆E}_{F}^{D}=-sign\left(n_{0}+N_{c}\right)\hbar v_{F}\sqrt{\pi \left|n_{0}+N_{c}\right|}+sign\left(n_{0}\right)\hbar v_{F}\sqrt{\pi \left|n_{0}\right|}$$
where *N_c_* represents the charges trapped in the defects at the a-Si:H/Gr interface which are released into Gr under illumination (by working at 1550 nm, all energy levels from *E_c_* down to 0.8 eV can be excited). The value of *N_c_* is completely determined by the generation rate of the charges released into Gr from the a-Si:H interface. Defining the flux of photons as $\psi =P/\left(h\nu \cdot A_{Gr}\right)$, where *P* (expressed in eV/s) is the constant incident optical power, *A_Gr_* is the illuminated active area of the a-Si:H/Gr interface in cm^2^ and hν is the photon energy, *N_c_* can be linked to the photon flux by means of this relation: $N_{c}=\tau \cdot \eta \cdot \psi$, where *τ* can be physically viewed as a carrier lifetime, while *η* is the conversion efficiency (adimensional), i.e., the number of charges trapped at the a-Si:H/Gr interface which are released into graphene per incident photon. The conversion efficiency depends on the interaction between photons and traps, and since the number of traps is fixed while the number of photons depends on the optical power intensity, a dependence of *η* on *P* is expected and made explicit in Equation (16).

The change in the Schottky barrier under illumination can be written as:(16)$$q∆\phi_{B}=-sign\left(n_{0}+N_{c}\right)\hbar v_{F}\sqrt{\pi \left|n_{0}+{sign(N}_{c})\cdot \tau \cdot \eta (P)\cdot \frac{P}{h\upsilon \cdot A_{Gr}}\right|}+sign\left(n_{0}\right)\hbar v_{F}\sqrt{\pi \left|n_{0}\right|}$$

Equation (16) shows a dependence on the efficiency-lifetime product, a complex parameter which can be influenced by many factors, and we can only speculate here that they might include the type, density, and energy level of the traps involved in the detection mechanism. Equation (16) replaced in Equation (12) gives the photocurrent:(17)$$I_{Ph}=A_{Gr}A_{N}^{*}T^{2}e^{-\frac{{q\phi }_{B}^{Gr}}{kT}}\left[e^{\frac{\hbar v_{F}}{kT}\left(-sign\left(n_{0}+N_{c}\right)\sqrt{\pi \left|n_{0}+{sign(N}_{c})\cdot \tau \cdot \eta (P)\cdot \frac{P}{h\upsilon \cdot A_{Gr}}\right|}+sign\left(n_{0}\right)\sqrt{\pi \left|n_{0}\right|}\right)}-1\right]$$
and the responsivity becomes:(18)$$R=\frac{I_{Ph}}{P}=\frac{A_{Gr}A_{N}^{*}T^{2}e^{-\frac{{q\phi }_{B}^{Gr}}{kT}}}{P}\left[e^{\frac{\hbar v_{F}}{kT}\left(-sign\left(n_{0}+N_{c}\right)\sqrt{\pi \left|n_{0}+{sign(N}_{c})\cdot \tau \cdot \eta (P)\cdot \frac{P}{h\upsilon \cdot A_{Gr}}\right|}+sign\left(n_{0}\right)\sqrt{\pi \left|n_{0}\right|}\right)}-1\right]$$

For our devices, the reduction in the Gr/c-Si Schottky barrier under illumination suggests that the charges released into Gr are electrons; therefore, *N_c_* can be considered with a negative sign. In addition, using Hall measurements we found that transferred Gr presents a natural P-type doping of 9 × 10^12^ cm^−2^ and that the P-type doping reduces to *n*_0_ = 3.5 × 10^12^ cm^−2^ after the deposition of the a-Si:H capping layer, as also reported in the literature [44]. Therefore, n_0_ can be considered with a positive sign and under the assumption that no doping inversion in graphene occurs by NIR illumination (|*N_c_*| < |*n*_0_|), Equation (18) reduces to:(19)$$R=\frac{I_{Ph}}{P}=\frac{A_{Gr}A_{N}^{*}T^{2}e^{-\frac{{q\phi }_{B}^{Gr}}{kT}}}{P}\left[e^{\frac{\hbar v_{F}}{kT}\cdot \left(-\sqrt{\pi \left|n_{0}-\tau \cdot \eta (P)\cdot \frac{P}{h\upsilon \cdot A_{Gr}}\right|}+\sqrt{\pi \left|n_{0}\right|}\right)}-1\right]$$

The experimental results shown in Figure 6, derived from Figure 5c at −21 V, are well fitted using Equation (19) by modelling the efficiency-lifetime carrier product as the power function *τη*(*P*) = *τη*_0_/*P^β^*. From the fitting procedure we extracted the following values: *τη*_0_ = 3.0 × 10^−9^ sW^β^, β = 0.79 and ${q\phi }_{B}^{Gr}=0.67$ eV (R-square of 0.99). In the fitting process, the following values have been used: *A_Gr_* = 0.0038 cm^2^, $A_{N}^{*}$ = 112 A/cm^2^K^2^, *T* = 300 K, *k* = 8.61 × 10^−5^ eV/K, ℏ = 6.58 × 10^−16^ eVs, $v_{F}={1.1\times 10}^{8}\;cm/s$, hυ=0.8 eV, and a capped Gr doping of *n*_0_ = 3.5 × 10^12^ cm^−2^. It is worth noting that the extracted potential barrier $\phi_{B}^{Gr}$ is in excellent agreement with what the theory predicts: *q*ϕ*_B_*_0_ = *E_g_* − *q*(ϕ*_Gr_* − χ_Si_) = 0.67 eV [40] (with the Si electron affinity *q*χ_Si_ = 4.05 eV [40], the graphene work function *q*ϕ*_Gr_* = 4.5 eV [45], and the Si bandgap *E_g_* = 1.12 eV [40]). The resulting *τη*(*P*) behavior is reported in the inset of Figure 6. The presence of a power law in the responsivity formula described by Equation (19) is not a novelty. It is widely reported in the literature that graphene photodetectors, whose detection mechanism is assisted by traps, provide a photocurrent that can be expressed by the simple power law *I_ph_* ∝ *P*^α^, where α is a non-unity exponent with values between zero and one as a result of the complex process of carrier generation, trapping, and recombination within the involved materials [5].

This dependence can easily be made explicit under the assumption |*N_c_*| << |*n*_0_|, and Equation (19) can be approximated providing the following photocurrent:(20)$$I_{Ph}≅\left[A_{N}^{*}T^{2}e^{-\frac{{q\phi }_{B}^{Gr}}{kT}}\cdot \left(\frac{\pi }{2}\cdot \frac{\hbar v_{F}}{kT}\cdot \frac{\tau \cdot \eta_{0}}{\sqrt{\pi n_{o}}\cdot h\nu }\right)\right]\cdot P^{1-\beta }$$

Therefore, by considering *I_ph_* = *G*·*P*, we have:(21)$$G≅\left[A_{N}^{*}T^{2}e^{-\frac{{q\phi }_{B}^{Gr}}{kT}}\cdot \left(\frac{\pi }{2}\cdot \frac{\hbar v_{F}}{kT}\cdot \frac{\tau \cdot \eta_{0}}{\sqrt{\pi n_{o}}\cdot h\nu }\right)\right]\cdot P^{-\beta }$$
where *G* represents the photo-gain which is based on the change in thermionic current due to the photo-induced graphene Fermi level modulation assisted by interfacial traps. Equation (21) clearly shows how the photo-gain *G* decreases by increasing the optical power *P* and how it strongly depends on the graphene Schottky barrier ${q\phi }_{B}^{Gr}$.

## 6. Conclusions

We assessed the behavior of near-infrared resonant cavity photodetectors based on graphene layers embedded between amorphous and crystalline silicon. The photodetectors are based on the Gr/c-Si/Al junction, which behaves like an MSM device and is able to support the current flow. The a-Si:H/Gr structure is not electrically active, but traps localized at the a-Si:H/Gr interface play a key role in the overall device performance, as evidenced by the dependence of the dark current on time and of the device responsivity on the incident optical power. The photodetection mechanism has been ascribed to the thermionic current modulation of the Gr/c-Si junction induced by the charge carriers released from traps, localized at the a-Si:H/Gr interface under illumination. The physics behind the operation of these devices was thoroughly derived, elucidated, and discussed, demonstrating a good agreement with the experimental results. At an optical power of 8.7 W (the minimum allowed by the equipment used), the responsivity of 27 mA/W at 1543 nm was obtained. Importantly, such a value can be further improved by using a higher-finesse cavity that may lead to an enhancement in the interaction between photons and interface traps, and consequently to an increase in the conversion efficiency. Furthermore, we have demonstrated that the responsivity of the PD can also be improved by lowering the incident optical power. Therefore, the better performance at low optical power makes our device useful in power monitoring within photonic integrated circuits, allowing non-invasive analysis in complex systems where the stabilization and control of several of the components is crucial [46]. Finally, the information extracted from the electrical and optical analysis herein reported can offer new insights for the development of NIR PDs based on the integration of silicon photonics with graphene technology. We believe that other 2D materials could be employed provided they have two main characteristics: (1) high optical transparency in the NIR spectrum and (2) the ability to form a Schottky junction with silicon. The former is necessary to consider the a-Si:H/2D material/c-Si three-layer hybrid junction as a single optical microcavity, while the latter allows the thermionic current which should be modulated to be generated.

## Figures and Tables

**Figure 1 nanomaterials-13-00872-f001:**
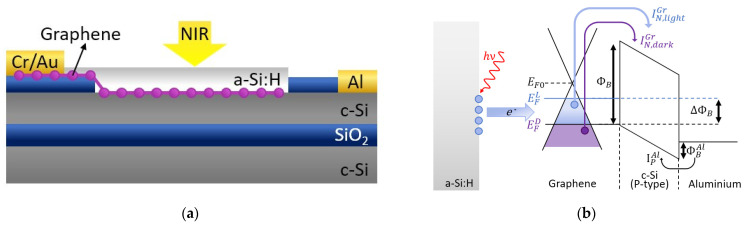
(**a**) Device sketch and (**b**) detection mechanism described by the energy band diagram for single-layer graphene under NIR illumination the traps at the a-Si:H/Gr interface release charge carriers into Gr, changing the Schottky barrier of the Gr/c-Si junction and hence the current flowing through the device.

**Figure 2 nanomaterials-13-00872-f002:**
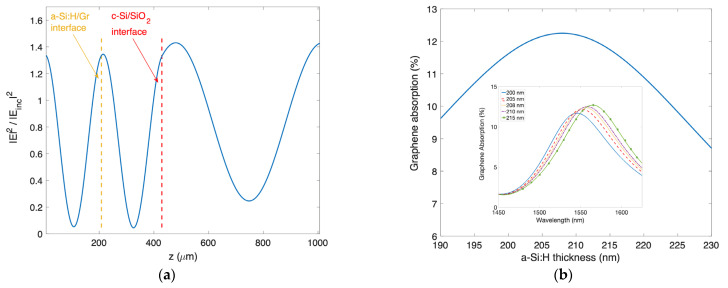
(**a**) Optical field distribution in the proposed photodetector as a function of position for a 208 nm-thick a-Si:H layer (z = 0 corresponds to the air/a-Si:H interface). (**b**) Theoretical Gr optical absorption for different thicknesses of the a-Si:H layer (inset: Gr absorption as a function of the wavelength for various a-Si:H thicknesses).

**Figure 3 nanomaterials-13-00872-f003:**
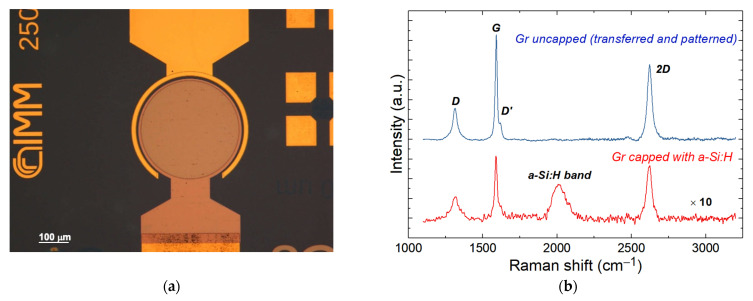
(**a**) Top view of the device taken by optical microscope. (**b**) Raman spectra of bare patterned graphene on silicon oxide (blue spectrum) and Gr capped with a-Si:H (red spectrum).

**Figure 4 nanomaterials-13-00872-f004:**
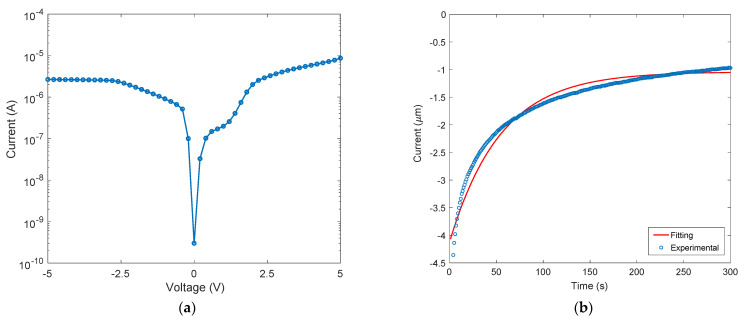
(**a**) I-V curve of the device and (**b**) time dependence of the measured dark current flowing through the device at a fixed voltage of −21 V.

**Figure 5 nanomaterials-13-00872-f005:**
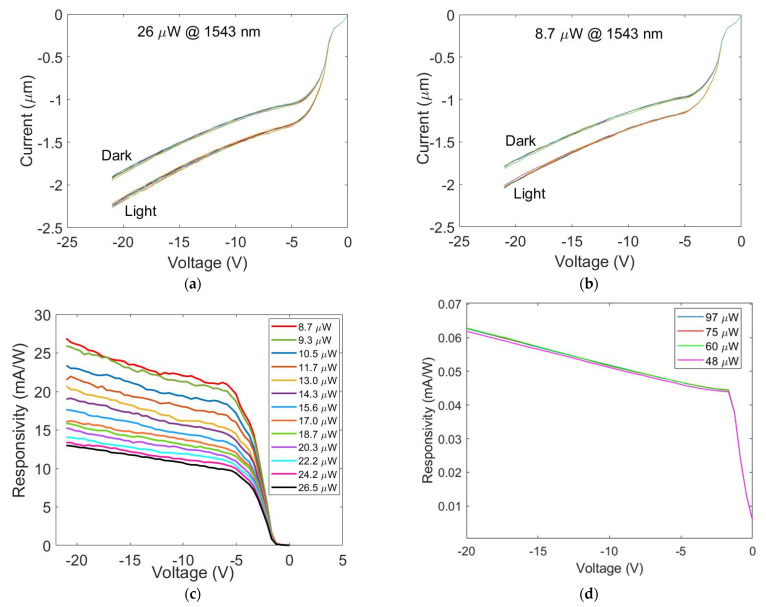
Dark current (*i_D_*) and current generated by NIR illumination (*i_L_*) measured for an optical power of (**a**) 26.5 µW and (**b**) 8.7 µW at 1543 nm. (**c**) Measured responsivity at 1543 nm vs. negative voltage applied for various optical power incidents on the device in Figure 1a. (**d**) Measured responsivity around 1550 nm vs. negative voltage applied for various optical powers incident on the Gr/c-Si Schottky junction before the a-Si:H deposition.

**Figure 6 nanomaterials-13-00872-f006:**
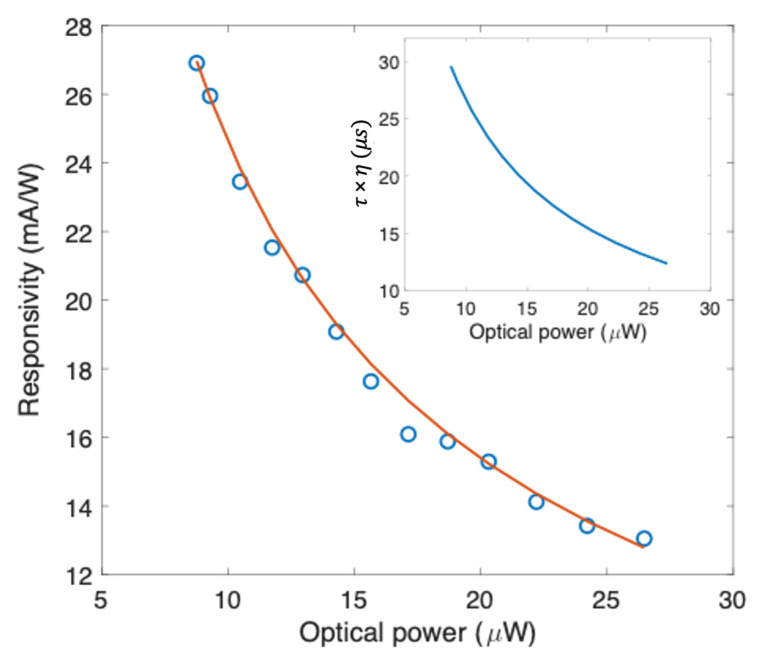
Responsivity at 1543 nm vs. optical power at −21 V (inset: efficiency-lifetime carrier product as function of the optical power).

**Figure 7 nanomaterials-13-00872-f007:**
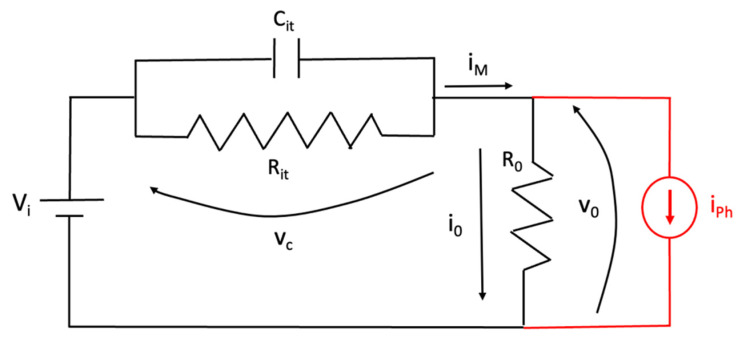
Electrical circuit schematizing the electrical behavior of the device provided with a current generator representing the photogenerated current (in red) under NIR illumination.

**Table 1 nanomaterials-13-00872-t001:** Frequency of D, G, and 2D phonon modes and ratio of the D and G peak intensity for bare and embedded graphene.

Sample	X_D_ (cm^−1^)	X_G_ (cm^−1^)	X_2D_ (cm^−1^)	I_D_/I_G_
Uncapped Gr	1312	1592	2623	0.29
Gr capped with a-Si:H	1314	1590	2625	0.41

## Data Availability

The datasets generated during the current study are available from the corresponding author on reasonable request.

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
