# Peer review of "The Physics behind the Modulation of Thermionic Current in Photodetectors Based on Graphene Embedded between Amorphous and Crystalline Silicon"

_nanomaterials, 2023, doi:10.3390/nano13050872_

Round 1

Reviewer 1 Report

The author repoet that the behavior of near-infrared resonant cavity photodetectors based on  graphene layers embedded between amorphous and crystalline silicon.

1. The author directly wrote his own conclusion, without explaining the significance of the research, nor explaining the comparison between his device and other people's results.

2.  The numbers and units directly need spaces in full text.

3.  Although the author gave a theoretical explanation of the experiment, as shown in Figure 7, we felt that it was not very clear for the reader to understand.

4.  The conclusion of the article does not need to be written in sections.

Reviewer 2 Report

In this article, authors have developed an infrared photodetector based on graphene. By sandwiching graphene between amorphous and crystalline silicon and using the charge trapping, de-trapping mechanism at the one of the interface authors has achieved excellent responsivity.

The article is very well written, and through characterization of materials and the devices were carried out, additionally it is supported by the detailed theoretical mechanism involved in the working of the device. Results and concepts will be a good reference for researchers working on graphene/2-d material-based devices and their integration with conventional microelectronics fabrication. Reviewer recommends the publication of the article, with some minor comments.   

·         Can their concepts be generalized for the other 2-D material-based devices?

·         Is there any threshold for the interface trap density to be effective in lowering the energy barrier? And the energy levels of the taps (eg. deep or shallow w.r.t graphene)?

·         Provided with the right values of the dielectric or material parameters would other materials than a-si will also work as a surface trapping layer?

·         Authors have mentioned vertical illumination, could author detail how the illumination is done from the a-si side or any other ways? 

Reviewer 3 Report

The authors study the physics behind the modulation of thermionic current in photodetectors based on graphene embedded between amorphous and crystalline silicon layers, proposing a mechanism to describe the increase of the photocurrent. The design and experimental results are discussed, and a model is proposed.

The manuscript relies, however, on formulas (equations 1, 13) and figures (Gr bandstructure in fig. 1(b)) valid for single-layer graphene, whereas the devices have 4-5 layers of graphene. It is well-known that the electronic and optical properties of graphene depend drastically on the number of layers; even bilayer graphene is different from monolayer graphene. Therefore, the authors should comment on the applicability of these formulas and figures in their work; more precisely to indicate the consequences on their model of applying not well-suited formulas.

In addition, whether the mechanism behind the photodetector is feasible, one should comment on the utility of such devices with a quite small responsivity.

The two points raised above should be properly addressed before the manuscript can be considered for publication

Reviewer 4 Report

In this work, the authors studied a vertically illuminated near-infrared photodetector based on graphene layer, which is physically embedded between crystal and silicon hydride layer. Under near-infrared illumination, the reported equipment showed an unexpected increase in thermionic current. This effect has been ascribed to the lowering of the graphene/crystalline silicon Schottky barrier as the result of an upward shift of the graphene Fermi level induced by the charge carriers released from traps localized at the graphene/amorphous silicon interface under illumination. The authors' findings provide new insights and highlight a new detection mechanism that can be used to develop near-infrared silicon photodetectors suitable for power monitoring applications. I believe that publication of the manuscript may be considered only after the following issues have been resolved.

1.       In order to better highlight the advantages of this work, the author needs to provide a table to compare related work.

2.       There should be a space between numbers and units, including in the chart.

3.       In the optical microscope picture in Figure 3, the upper scale needs to be added.

4.       How is graphene prepared in the device in this work? In situ growth or? The author needs to introduce it in the Device concept and design section.

5.       The introduction can be improved. The articles related to some applications of graphene materials should be added such as Sensors 2022, 22, 6483; ACS Sustain. Chem. Eng. 2015, 3, 1677–1685; Diamond & Related Materials 128 (2022) 109273; Talanta 2015, 134, 435–442.

6.       Please check the grammar and spelling mistakes of the whole manuscript.

Round 2

Reviewer 1 Report

The author has revised the manuscript according with the comment. I advise to accept it.

Reviewer 3 Report

The manuscript can now be published. The authors addressed the issues raised in the review process quite satisfactorily

Reviewer 4 Report

Accept in present form.